# Development of a Treatment Planning Framework for Laser Interstitial Thermal Therapy (LITT)

**DOI:** 10.3390/cancers15184554

**Published:** 2023-09-14

**Authors:** Yash Lad, Avesh Jangam, Hayden Carlton, Ma’Moun Abu-Ayyad, Constantinos Hadjipanayis, Robert Ivkov, Brad E. Zacharia, Anilchandra Attaluri

**Affiliations:** 1Department of Mechanical Engineering, School of Science, Engineering, and Technology, The Pennsylvania State University Harrisburg, Harrisburg, PA 17057, USA; 2Department of Radiation Oncology and Molecular Radiation Sciences, The Johns Hopkins University School of Medicine, Baltimore, MD 21231, USA; 3Department of Neurological Surgery, University of Pittsburgh School of Medicine, Pittsburgh, PA 15213, USA; 4Department of Oncology, Johns Hopkins University School of Medicine, Baltimore, MD 21287, USA; 5Department of Mechanical Engineering, Whiting School of Engineering, Johns Hopkins University, Baltimore, MD 21218, USA; 6Department of Materials Science and Engineering, Whiting School of Engineering, Johns Hopkins University, Baltimore, MD 21218, USA; 7Department of Neurosurgery, Pennsylvania State Health, Hershey, PA 17033, USA

**Keywords:** LITT, GBM, treatment panning, bioheat transfer simulation, GUI

## Abstract

**Simple Summary:**

Simulation-based tools provide a platform for planning LITT procedures. By incorporating patient-specific imaging data, such as MRI or CT scans, into simulations, the treatment planning tool can generate a three-dimensional representation of the tumor and surrounding brain tissue. This enables surgeons to plan the optimal placement and trajectory of the laser probe and control energy delivery to ensure safety and optimize tumor thermal dose coverage.

**Abstract:**

Purpose: Develop a treatment planning framework for neurosurgeons treating high-grade gliomas with LITT to minimize the learning curve and improve tumor thermal dose coverage. Methods: Deidentified patient images were segmented using the image segmentation software Materialize MIMICS©. Segmented images were imported into the commercial finite element analysis (FEA) software COMSOL Multiphysics© to perform bioheat transfer simulations. The laser probe was modeled as a cylindrical object with radius 0.7 mm and length 100 mm, with a constant beam diameter. A modeled laser probe was placed in the tumor in accordance with patient specific patient magnetic resonance temperature imaging (MRTi) data. The laser energy was modeled as a deposited beam heat source in the FEA software. Penne’s bioheat equation was used to model heat transfer in brain tissue. The cerebrospinal fluid (CSF) was modeled as a solid with convectively enhanced conductivity to capture heat sink effects. In this study, thermal damage-dependent blood perfusion was assessed. Pulsed laser heating was modeled based on patient treatment logs. The stationary heat source and pullback heat source techniques were modeled to compare the calculated tissue damage. The developed bioheat transfer model was compared to MRTi data obtained from a laser log during LITT procedures. The application builder module in COMSOL Multiphysics© was utilized to create a Graphical User Interface (GUI) for the treatment planning framework. Results: Simulations predicted increased thermal damage (10–15%) in the tumor for the pullback heat source approach compared with the stationary heat source. The model-predicted temperature profiles followed trends similar to those of the MRTi data. Simulations predicted partial tissue ablation in tumors proximal to the CSF ventricle. Conclusion: A mobile platform-based GUI for bioheat transfer simulation was developed to aid neurosurgeons in conveniently varying the simulation parameters according to a patient-specific treatment plan. The convective effects of the CSF should be modeled with heat sink effects for accurate LITT treatment planning.

## 1. Introduction

Laser interstitial thermal therapy (LITT) is a minimally invasive surgical technique used to treat neurological disorders, such as epilepsy, and brain tumors, such as glioblastoma (GBM) [1,2,3]. The effectiveness of LITT depends on the precise insertion of a laser probe into the tissue with controlled application of laser energy to ablate the target tissue without damaging nearby healthy tissue [4]. Real-time magnetic resonance (MR) or computed tomography (CT) imaging guides procedures to ensure precise placement and monitoring of the laser probe. Stereotactic magnetic resonance-guided LITT (MRgLITT) has gained prominence in treating epilepsy and recurrent GBMs since its approval by the US Food and Drug Administration in 2007 [2,3]. MRTi enables real-time in situ temperature monitoring of brain tissue for MRgLITT [3].

Even with MRgLITT to assist the placement of the laser probe and monitor tissue temperature, outcomes depend on operator skill and experience to avoid off-target thermal damage while ensuring effective ablation of the target tissue [5]. Complexity in stereotactic laser placement and lack of treatment planning further increases the time necessary for an operator to achieve proficiency [6,7]. LITT treatment requires integrated trajectory planning for placement of the probe and bioheat transfer-based simulations for thermal dose planning. Trajectory planning tools are well established in the clinic [8,9,10,11]. Bioheat transfer simulations and LITT treatment planning tools reported in the literature help to understand the heat transfer dynamics and optimize treatment planning parameters. However, these previous efforts have produced information for engineers and medical physicists, without explicitly considering the end-user [12,13,14,15,16]. Currently, there is no LITT treatment planning framework to aid neurosurgeons when they plan thermal dose delivery with LITT. This leads to highly variable, and user-dependent lesion coverage with diminished predictability of outcomes for patients [6,7]. A LITT treatment planning framework that enables neurosurgeons to create robust, reliable, and reproducible LITT plans quickly using mobile computing will reduce the learning curve, improve the lesion coverage and quality control [6,7,9,10,11,17,18]. Ultimately, we can expect that widespread adoption of such tools will manifest with improved outcomes for GBM patients.

Margret et al. demonstrated that poor laser positioning can result in inadequate ablation or adverse events such as hemorrhage or unintended hyperthermia in normal brain tissue [6,7]. They noted that improvements correlated with experience and use of a trajectory planning tool for accurate placement of the probe [6,7]. This has motivated development of an accessible and precise treatment planning tool that combines stereotactic placement of the laser catheter to enhance successful LITT treatment. Mitchell et al. demonstrated a treatment-planning approach for MRgLITT in a heterogeneous tissue model [16]. They compared the performance evaluation criterion for optimization of model parameters for homogeneous and heterogeneous tissue models. However, their model does not predict ablation outcomes during treatment planning to evaluate relative merits of alternative possible trajectories [19].

Treatment tools developed to aid LITT have improved the control of recurrent tumors previously treated with radiotherapy [6,20]. The methodology employed by our proposed framework involves the utilization of imaging data, such as MRI or CT, to generate a three-dimensional representation of the tumor and adjacent brain tissue [3,19]. The thermal properties of the tissue are modeled to predict temperature changes that are likely to occur [21]. The objective of the planning tool is to achieve a temperature that is adequate to ablate the tumor while minimizing the temperature increase in the surrounding healthy brain tissue [22]. However, it is important to consider a 2 mm margin surrounding the tumor to decrease the likelihood of local recurrence [5]. Many GBM tumors are present near the CSF within the ventricular system [23]. In such cases, a reduction in temperature can occur during LITT owing to the heat sink effects of the CSF and surrounding tissue. The impact of heat sinks produced by the CSF, ventricles, posterior cerebral artery, and suprasellar and ambient cisterns on the local temperature during LITT has been identified in the literature [24,25,26], but has not been studied.

Another challenge encountered with LITT is adequate thermometry. Although MRTi has benefits, MR-guided LITT devices may overheat tissue owing to imperfections in the MRTi data, according to a 2019 FDA letter to healthcare providers [27]. To compensate, the treatment planning tool can incorporate sophisticated temperature modeling algorithms that simulate the heat distribution within the target tissue during LITT procedures. These models can consider various factors such as tissue properties, blood flow, and heat dissipation, providing a more accurate prediction of temperature changes. By accounting for imperfections in MRTi data, the tool can compensate for any discrepancies and provide reliable temperature estimations [24].

To optimize the preoperative planning of LITT for GBM, we sought to develop a mobile platform-based framework for LITT treatment planning that will aid in determining the treatment parameters to approximate lesion coverage. Current treatment planning strategies lack the ease of varying simulation parameters and acquiring thermal damage contours according to user-defined criteria to predict ablation outcomes and trajectories. To overcome this limitation, we proposed a treatment planning workflow for LITT using various strategies to improve the lesion coverage. We used bioheat transfer simulations to understand the heat conductivity and perfusion with neuroanatomical variations.

A cylindrical laser probe was modeled with a constant beam diameter to accommodate multiple probe tips from manufacturers for different cooling strategies in the framework [28,29]. Based on a previous study, we initially modeled the CSF as a solid material with a thermal conductivity of 6.2 [W/m·K], which is ten-fold higher than the standard value [30]. We also modeled the CSF as a fluid with enhanced conductivity owing to convection. The heat sink effects of the CSF can be studied better using this modeling approach. We utilized the Arrhenius kinetics model with pulsed laser power. A pullback heat source mechanism that is approximately identical to that used for patients in clinical settings was modeled. We followed the conventional approach by converting DICOM images to 3D geometries and importing them into the FEA software for modeling. For future clinical implementation, we envision bioheat transfer simulations, including the location of the tumor corresponding to the CSF ventricles, which will be modeled by the medical physics team using FEA software. The GUI for the treatment planning framework will be developed using the application builder module in COMSOL Multiphysics©. The surgical parameters will be conveniently varied by neurosurgeons with the aid of the GUI tool, and simulations will be performed in the background. The simulation results will be displayed on a mobile platform, aiding neurosurgeons in treatment decision making. The limitations of tissue overheating observed in MRgLITT can be improved with the lesion coverage approximations provided by this treatment planning tool.

## 2. Materials and Methods

### 2.1. Medical Image Segmentation

Digital Imaging and Communication in Medicine (DICOM) images obtained from MR and CT scans were segmented into 3D models using the segmentation software Materialize MIMICS Innovation Suite© (Plymouth, MI, USA) [31]. The individual 3D portions were meshed to appropriate tolerances in 3-Matic software and then imported into the FEA software (COMSOL Multiphysics©, Burlington, MA, USA). Figure 1 shows the workflow of the image segmentation to 3D model conversion.

### 2.2. Importing the 3D Model into FEA Software

A schematic of the simulation workflow is shown in Figure 2. Segmented 3D features assembled from 2D DICOM images were imported into COMSOL Multiphysics©. Spatial registrations were maintained using the segmentation software to ensure that the imaging components were correctly assembled when they were imported into the FEA software. For the present study, we considered the skull, CSF general, brain (white matter and grey matter averaged), CSF ventricles, and tumor.

### 2.3. Assigning Material Properties

The material properties assigned to individual domains are addressed in Table 1.

### 2.4. Modeling the Bioheat Transfer Equation with Time Dependent Study

Heat transfer was modeled using a modified version of Penne’s time-dependent partial differential equation (PDE), which accounts for blood perfusion, lasers, and metabolic heat sources [36,37].
(1)$$\rho_{i}C_{p,i}\frac{\partial T}{\partial t}-\nabla (k_{i}\nabla T)=Q_{perf}+Q_{Laser}+Q_{met}$$
(2)$$Q_{perf}=\rho_{b}C_{p,b}\omega_{b}\left(T_{b}-T\right)$$

The tissue was treated as a solid medium, where k is the thermal conductivity, T is the tissue temperature, $C_{p}$ is the heat capacity, and ρ is the density, as listed in Table 1 [36,37]. In Equation (2), $\rho_{b}$ is the density of blood, $C_{p.b}$ is the blood heat capacity, $\omega_{b}$ is the blood perfusion rate, and T_b_ is the blood temperature. From Equation (1), $Q_{Laser}$ is the external heat source (laser in this case), and $Q_{met}$ is the metabolic heat generation [37]. The subscript ‘i’ denotes the respective domains under consideration. For instance, i = skull/brain/tumor/csf when a particular domain in the bioheat transfer study is calculated.

### 2.5. Modeling Temperature Distribution

The initial study commenced with an analysis of the temperature distribution within the lesion caused by the heat source, which ranged from 37 °C to 57 °C. The term $\frac{\partial T}{\partial t}$ from bioheat transfer Equation (1) was used to plot the temperature distribution. The temperature distribution was divided into different zones based on temperature ranges, including 37 °C to 43 °C, 43 °C to 47 °C, 47 °C to 57 °C, and 57 °C and above.

### 2.6. Modeling Thermal Damage Dependent Blood Perfusion

We modeled two perfusion models, namely, the constant perfusion model based on Pennes’ bioheat equation, and the modified Arrhenius thermal damage-dependent perfusion model proposed by He et al. [38]. The modeling of this approach was performed because the Medtronic Visualase system utilizes Arrhenius damage in clinical settings [39]. For the constant perfusion model, the standard value of blood perfusion, $\omega_{b,tumor}=222.3\;[\frac{\frac{mL}{min}}{kg}]$ is used in the following equation:(3)$$\rho_{tumor}C_{p,tumor}\frac{\partial T}{\partial t}-\nabla \left(k_{tumor}\nabla T\right)=\rho_{b}C_{p,b}\omega_{b,tumor}\left(T\right)+Q_{Laser}+Q_{met}$$

For thermal damage-dependent perfusion, the relationship between blood perfusion and temperature as well as the degree of microvascular stasis (DS) over time was established using the Arrhenius model. The dimensionless value DS can be expressed as shown in Equation (4) [38,40,41].
(4)$$DS=1-\exp\left(-A\int_{0}^{t}e^{-\frac{E_{a}}{RT\left(\tau \right)}}\right)d\tau$$
with
(5)$$\omega_{b}\left(T\right)=\omega_{b,i}(1-DS)$$

In Equations (4) and (5), A is the frequency or pre-exponential factor [1/s], $E_{a}$ is the activation energy [J/mol], R is the universal gas constant [J/K·mol]; T(τ) is the absolute tissue temperature as a function of time; and $\omega_{b,i}$ is the initial blood perfusion rate [$\frac{mL}{min}/kg$]. The degree of vascular stasis DS varies between 0 and 1, that is, no vascular damage or complete vascular damage, respectively [38,40]. The values of A and $E_{a}$ are given in Table 2.
(6)$$\omega_{b}\left(T\right)=\left\{\begin{array}{c} \omega_{bi}\left(30DS+1\right),\;\left(DS\leq 0.02\right)\\ \omega_{bi}\left(-13DS+1.86\right),\;\left(0.02<DS\leq 0.08\right)\\ \omega_{bi}\left(-0.79DS+0.884\right),\;\left(0.08<DS\leq 0.97\right)\\ \omega_{bi}\left(-3.87DS+3.87\right),\;\left(0.97<DS\leq 1.0\right) \end{array}\right.$$

When the degree of stasis was minimal (DS ≤ 0.02), the perfusion was minimal. As the degree of stasis increases (0.02 < DS ≤ 0.08), perfusion decreases gradually at first and then more quickly (0.08 < DS ≤ 0.97) until it reaches zero (0.97 < DS ≤ 1.0) [41].

### 2.7. Calculating CEM43 Thermal Dose Volume

In this case, the cumulative equivalent minutes at 43 °C were modeled to calculate the time-dependent temperature changes and thermal dose volume. CEM43 was modeled for 10 min in this study (Monteris Neuroblate [43,44,45]) and is given by Equation (7) [41]:(7)$$CEM43=\int_{t=0}^{t=final}B^{\frac{43{}^{\circ}C-T\left(x,y,t\right)}{1{}^{\circ}C}}$$

The constant B has been observed to have a value of 0.5, T > 43 °C, and a value of 0.25 for temperatures, T ≤ 43 °C [38,40]. The CEM43 calculations were performed post-simulations, and the contours were plotted accordingly.

### 2.8. Modeling CSF and CSF Ventricles

To examine the difference in thermal deposition due to the heat sink effect of the CSF, we modeled two cases.

CSF and CSF ventricles modeled as solids with high thermal conductivity, 6.2 [W/m·K] andCSF and CSF ventricles modeled as fluids with convectively enhanced conductivity.

In the first case, we modeled the CSF and CSF ventricles with a thermal conductivity ten-fold higher than the standard value [30]. The thermal conductivity of the CSF was obtained from Schooneveldt et al., and was used as a general reference with a value of 6.2 [W/(m·K)] [30]. The bioheat transfer equation can be written as
(8)$$\rho_{csf}C_{p,\mathrm{csf}}\frac{\partial T}{\partial t}-\nabla (k_{\mathrm{csf}}\nabla T)=Q_{\mathrm{Laser}}$$

In the second case, we modeled the CSF and CSF ventricles as fluids with convectively enhanced conductivities. The thermal conductivity was assumed to be 0.62 [W/(m·K)] [30]. Equation (11) was utilized to compute the Nusselt number, Nu, with the assumption that the angle of inclination from the vertical (θ) is 30°, while Gr and Pr represent the Grashof and Prandtl numbers, respectively [46]. The Grashof number was calculated using Equation (10), where Re is the Reynolds number [47]. The properties of the fluid, along with the numerical values for the Pr, Re, Gr, and Nu, are detailed in Table 3:(9)$$\mathrm{Pr}=C_{p}\mu /k$$
(10)Gr=Re/Pr
(11)$$Nu={0.67\;[Gr*Pr*\cos\left(\theta \right)]}^{0.25}$$
(12)q=−kNu∇T

Considering CSF as a fluid with convectively enhanced conductivity, the blood perfusion term in Equation (1) becomes zero. Therefore, Equation (8) can be written as,
(13)$$\rho_{csf}C_{p,csf}\frac{\partial T}{\partial t}-\nabla (k_{csf}Nu\nabla T)+\rho_{csf}C_{p,csf}u\nabla T=Q_{Laser}$$

### 2.9. Laser-Tissue Interaction

The laser applicator is a disposable cylindrical silica fiber probe enclosed in a saline-cooled polycarbonate catheter [49]. The laser applicator was secured to a threaded plastic bone anchor, which was mounted on the skull using a battery-powered hand drill [49]. The laser probe was modeled as a cylinder with a radius 0.7 mm and height 100 mm. Laser probe insertion was modeled from the top at a specific angle of 5° relative to the vertical axis based on neuronavigation.

Equations (14) and (15) were used to mathematically model the laser heat source as deposited beam power. The Gaussian distribution f(O,e) is characterized by the beam origin point O, and the beam orientation, e. The standard deviation of the Gaussian profile is represented by σ and d is the beam diameter. The value of q $\left[W/m^{2}\right]$ was evaluated using Equation (12), which was subsequently utilized to evaluate the heat source value $Q_{Laser}$ $\left[W/m^{3}\right]$. The variable $P_{Laser}$ [W] represents the laser power in this study.
(14)$$F\left(O,e\right)=\frac{1}{2\pi \sigma^{2}}\exp\left(-\frac{d^{2}}{2\sigma^{2}}\right),d=\frac{\left\|e.(x-O)\right\|}{\left\|e\right\|}$$
(15)$$-n.q=P_{\mathrm{Laser}}.f\left(O,e\right).\frac{\left|e.n\right|}{\left\|e\right\|}$$

### 2.10. Defining Idealized Computational Model

To investigate the behavior of the model with respect to the material properties and boundary conditions, it is imperative to create a standardized model and validate the outcomes through analytical computations. The laser probe was subjected to a constant laser power of 9 Watts for 150 s during the initial phase of the simulation. To assess the temperature distribution, three temperature measuring point probes were placed for temperature measurement in the FEA software, as shown in Figure 3. These probes were strategically placed at multiple locations in the tumor, surrounding the laser probe in such a way that the first probe was positioned 1 mm away from the laser applicator corresponding to point one, the second probe was positioned 4 mm away from the laser probe corresponding to point two, and the third probe was positioned 7 mm away from the laser applicator corresponding to point three, just near the boundary of the tumor, as shown in Figure 3. The idealized geometry was created to comply with the ASME VVUQ40 standards for computational modeling and to verify the results against the analytical solution [50].

### 2.11. Modeling Convective Cooling Induced by the CSF Based on Tumor Location

To observe the difference in temperature deposition based on the tumor location, we simulated two distinct location scenarios for tumors. In this approach, the laser power was arbitrarily increased from 9 Watts to 15 Watts. The first scenario involved exposing the tumor positioned 3 mm from the CSF ventricle, as shown in Figure 4A. The second scenario involved positioning the tumor between the two CSF ventricles, commonly referred to as Butterfly Gliomas, as shown in Figure 4B.

### 2.12. Modeling Stationary Heat Source with Pulsed Laser Power

After verifying the simulation results of the idealized model with the analytical solution, we acquired a Graphical Interchange Format (gif) file of the laser log from the LITT surgery, and the data from this file were used to model the pulses in the simulation. According to the laser log, the surgery lasted for 794 s, involving a 25% duty cycle. The laser probe was fully implanted within the tumor, with a laser diffusing tip measuring 5 mm in length. Throughout the process, the heat source was maintained stationary.

### 2.13. Modeling Pullback Laser Probe with Pulsed Laser Power

To maximize lesion coverage, we modeled a pullback heat source in which the 5 mm-long laser diffusing tip was substituted with a 15 mm-long laser diffusing tip. The heat source was modeled using a piecewise function, with the laser probe modeled using the static meshing approach in FEA software within the tumor. The pullback of the heat source was varied in the range of 15 mm. During the simulations, the heat source was subjected to alternating periods of being turned off and maintained stationary. Specifically, the heat source was pulled back for a duration when the heat pulse was turned off and remained stationary for a duration when the heat pulse was turned on. The process was performed until the heat source reached the proximal end of the tumor, with a gap of approximately 2 mm from the boundary of the tumor. The study involved initiating the movement of the heat source from the distal end of the tumor and halting it upon reaching the proximal end. The temperature measuring point probes were modeled using a static approach in the FEA software. Temperature measuring point probes can be placed around the tumor according to the user’s will to obtain accurate temperature distribution data and control the temperature around the tumor boundary.

### 2.14. Creating a Graphical User Interface (GUI)

Figure 5 depicts the interface of the treatment planning framework, which is designed to be user friendly. The graphical user interface (GUI) displays a range of visualizations, including mesh geometry, temperature distribution, thermal damage, laser pulse, and temperature patterns at multiple probes resulting from the pulse. The GUI was deployed using the COMSOL Multiphysics© application builder module. The segmented 3D geometry was imported by the medical physics team according to patient-specific data, and the bioheat transfer equation and coupled Multiphysics were set in COMSOL. Once the model is prepared in COMSOL Multiphysics©, patient-specific data can be fed by the neurosurgeon by adjusting the slider bars on mobile platforms, such as iPads [9,10,11]. Simulations will be performed on the server, and the results were displayed on the mobile platform. The parameters of the surgery, including duration, laser power, and pulse, can be adjusted by the user to observe the initial effects of the thermal dose. This tool can help determine the optimal probe location and angle for complex surgical procedures. The collected data can be used to analyze blind spots that may occur during a surgical procedure.

## 3. Results

### 3.1. Stationary Heat Source with Constant Laser Power

Figure 6 shows the simulation results of the idealized model. The maximum temperature observed during the procedure was approximately 70.69 °C over a duration of 150 s. The application of constant laser power produced a linear increase in temperature. Additionally, owing to a constant blood perfusion value, there was a rapid increase in temperature in relation to the laser power. The results showed conformity with the analytical solution, verifying the computational modeling in accordance with the ASME VVUQ40 standards [32].

### 3.2. Convective Cooling Induced by CSF and CSF Ventricles

The results in Figure 7 predicted higher calculated maximum temperatures when the simulated tumor was located farther from the CSF ventricles, whereas lower temperature deposition was predicted when the tumor was modeled closer to the ventricles. The temperature profile of tumors located farther away from the ventricles exhibited slight nonlinearity due to blood perfusion.

### 3.3. Stationary Heat Source with Pulsed Laser Power

Figure 8 shows the conformity of the predicted temperatures with the model laser pulse at different time intervals. Because the probe was stationary throughout the simulation, the temperature deposition occurred only in the region surrounding the laser tip. The maximum temperature recorded was approximately 71.68 °C.

### 3.4. Pullback Heat Source with Pulsed Laser Power

The movement of the heat source corresponding to the laser pulse is illustrated in Figure 9A. Compared to the stationary probe technique, the temperature predicted by this method was relatively high. The pullback technique estimated 70% thermal damage to the tumor, with a total volume of 3011.1 mm^3^. The temperature at point three, i.e., on the tumor boundary, was predicted to be below the necrosis temperature of 57 °C, according to the results displayed in Figure 9B. The pullback heat source results were captured using animation plots in COMSOL Multiphysics©. Figure 10 illustrates the correlation among temperature distribution, thermal damage-dependent blood perfusion, and CEM43 outcomes at 10 min, with data from the MRTi laser log.

## 4. Discussion

The bioheat transfer simulations predicted the temperature changes and thermal effects during the LITT. By considering the thermal properties of the tissue, including the conductivity and perfusion, the simulation framework can be used to estimate the temperature distribution and extent of tissue ablation [21]. This information is crucial for optimizing treatment parameters and minimizing the risk of off-target thermal damage to healthy brain tissues [5]. Simulation-based tools can aid in optimizing the lesion coverage during LITT. By modeling the tumor and surrounding tissue, including the impact of CSF and heat sink effects, this framework can help identify potential areas of incomplete ablation and guide neurosurgeons in adjusting the treatment parameters to ensure adequate coverage of the tumor. This optimization can potentially reduce the likelihood of tumor recurrence and improve treatment outcomes.

However, simulation models used in treatment planning tools often involve simplifications and assumptions to make the calculations computationally feasible. These simplifications may not fully capture the complex and dynamic nature of brain tissue and its response to thermal ablation. The accuracy of the simulation results is highly dependent on the accuracy of the input parameters and assumptions made during the modeling process. The precise values of tissue properties, such as thermal conductivity, perfusion rates, and heat transfer coefficients, can be challenging to measure or accurately estimate [21]. Inherent uncertainties in these properties can introduce errors in simulation-based treatment planning, thereby affecting the reliability and accuracy of the predicted outcomes. It is essential to consider these uncertainties and their potential impact on the planning process. In a study conducted by Mitchell et al., they improved the accuracy of MRgLITT outcome prediction by modeling four different tissue types with independent optical properties [16]. However, their modeling approach does not predict ablation outcomes during treatment or pre-treatment planning. To overcome this limitation, we developed a framework which provides the ablation outcomes in three different ways i.e., T > 57 °C, CEM43 t10 and Arrhenius damage of the lesion [39,43,44,45]. It is also possible to evaluate possible trajectories with the help of this tool to identify the maximum lesion coverage without off-target thermal deposition [19]. Previously developed LITT treatment planning tools aid engineers and medical physicists in terms of trajectory planning and lesion coverage, without explicitly considering the end-user [12,13,14,15,16].

### Proposed Workflow for LITT Treatment Framework

To overcome the above-mentioned limitations, we propose a workflow for planning LITT treatment as shown in Figure 11. According to this workflow, neurosurgeons can change the neural navigation plan based on MRI and CT scans with the help of a GUI. Simulations will be performed in the background after setting the required parameters, such as the probe location, laser power, laser pulse, and duration of the treatment were adjusted. The results will be displayed on a mobile platform [9,10,11,17,18]. The predicted results can be examined by neurosurgeons, and the treatment parameters can be used to determine the optimal solution. Once the specific patient’s optimal treatment parameters are defined, clinical treatment can be performed by neurosurgeons.

The LITT treatment comprises several consecutive temperature pulses [51]. Each pulse contained a heating and cooling portion, which could be fitted to analytical model(s) and used to extract a wealth of thermal information. For example, in a recent study by Desclides et al., an initial, non-destructive temperature pulse was used at the beginning of the study to estimate the thermodynamic parameters of the tissue, which were then implemented into a PID control algorithm [52,53]. By empirically measuring the thermal properties of the tissue, the number of simulation assumptions decreases, which improves the accuracy of the simulation and treatment planning utility. Additional efforts have utilized the nonadiabatic Box–Lucas model to quantify the expended thermal energy during pulsed heating and actively monitor the evolution of the convective heat loss during the measurement [54]. Each temperature pulse serves as a snapshot of the thermal transport occurring within the tissue at a given time interval, and this information should be utilized to tune the LITT treatment in situ. Accumulated tissue damage during treatment significantly affects heat removal via perfusion, as seen with the Arrhenius relation, which manifests as a dampened cooling response. Monitoring this response provides biologically relevant feedback that would enable active adjustment of laser power to improve treatment outcomes.

To implement the proposed treatment plan, it is imperative to develop a control strategy that considers both the probe movement and thermal dose deposition using a robust controller. Our future studies will utilize the Model Predictive Control (MPC) algorithm to establish a reliable control mechanism for LITT. In this control strategy, the dynamic matrix of the MPC is expanded to a Multi-Input Multi-Output (MIMO) system. In this MIMO structure, the temperatures of the tumor center and its boundary are considered as the controller outputs, whereas the probe location and laser power are used as inputs for the MIMO approach. The dynamic matrix used in this controller will be developed from open-loop tests of the thermal probe after being analyzed using the autoregression modeling technique. The nondestructive temperature pulse at the beginning of the study and pulsed laser heating with real-time MRTi will provide feedback to the MPC. In the MPC approach, the error vector is calculated as the difference between the setpoint vector and prediction vector of the controlled variable (tumor temperature). The control moves of the MPC controller are calculated by minimizing the cost function using a simple least square of the error vectors. The calculated controlled move is then sent to the electrical heater to raise the temperature of the tumor until it reaches the setpoint reference.

The integration of neuronavigation with the proposed LITT treatment planning framework is critical for a seamless clinical workflow. This integration helps mitigate the risks associated with off-target thermal doses owing to the heat sink effects caused by the CSF ventricles. Future studies are needed to identify optimal patient-specific treatment parameters, such as laser positioning and laser pulsing parameters, and develop a robust pull-back control system to safeguard surrounding healthy tissues from potential damage during LITT.

## 5. Conclusions

In this study, we developed a LITT treatment planning framework with GUI application in COMSOL that empowers neurosurgeons to autonomously adjust patient-specific surgical parameters on a mobile platform and visualize lesion coverage during LITT execution of the plan. It is important to model heat sink effects caused by the presence of the fluid structures such as CSF ventricles. To ensure reliability, the developed framework requires verification and validation in accordance with the ASME VVUQ40 standards before it can be implemented in clinical trials.

## Figures and Tables

**Figure 1 cancers-15-04554-f001:**
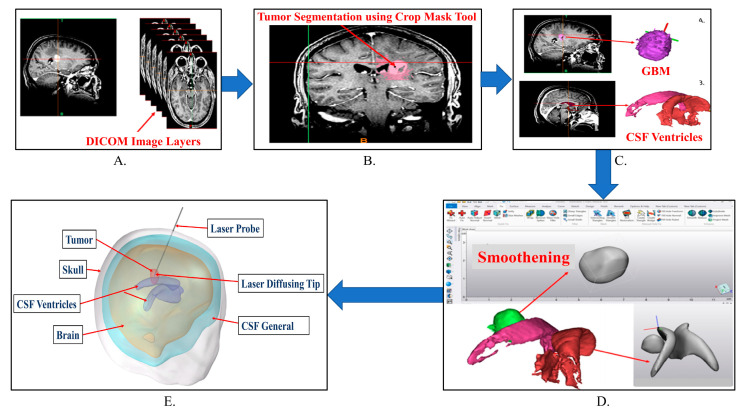
Steps for conversion of DICOM images to the STL file. (**A**) DICOM image representation in the form of different layers. (**B**) Thresholding to segment area of interest in MIMICS© using the crop mask tool to zoom and select the affected tissue and ignoring the rest of the selection. (**C**) Separating two regions in the segmentation using the split mask tool to get a 3D preview of GBM and CSF ventricles. (**D**) Smoothening operation on GBM and CSF ventricles in 3-Matic© software and conversion into STL file. (**E**). 3D assembled human head model imported to FEA software.

**Figure 2 cancers-15-04554-f002:**
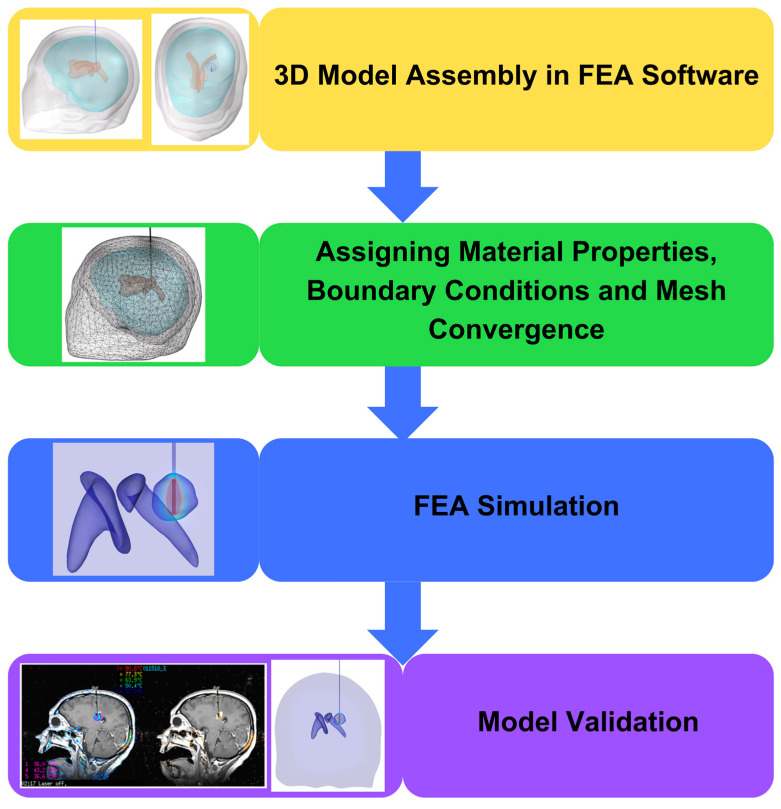
Simulation Workflow: The FEA software imports 3D parts and assembles them into a unified geometry. The model is prepared for simulations by assigning material properties and boundary conditions, followed by conducting mesh convergence. The study employs Finite Element Analysis (FEA) simulations to investigate specific scenarios using the bioheat transfer equation and a time-dependent approach. The American Society of Mechanical Engineers (ASME) standards can be used to verify and validate simulation outcomes.

**Figure 3 cancers-15-04554-f003:**
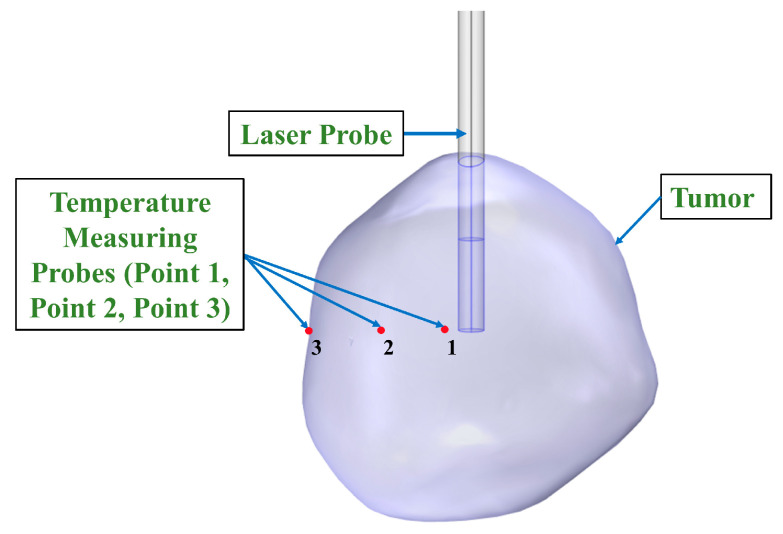
Location of the temperature measuring probes with respect to the laser applicator in the tumor. Point one is placed 1 mm away, point two is placed 4 mm away and point three is placed 7 mm away from the laser probe.

**Figure 4 cancers-15-04554-f004:**
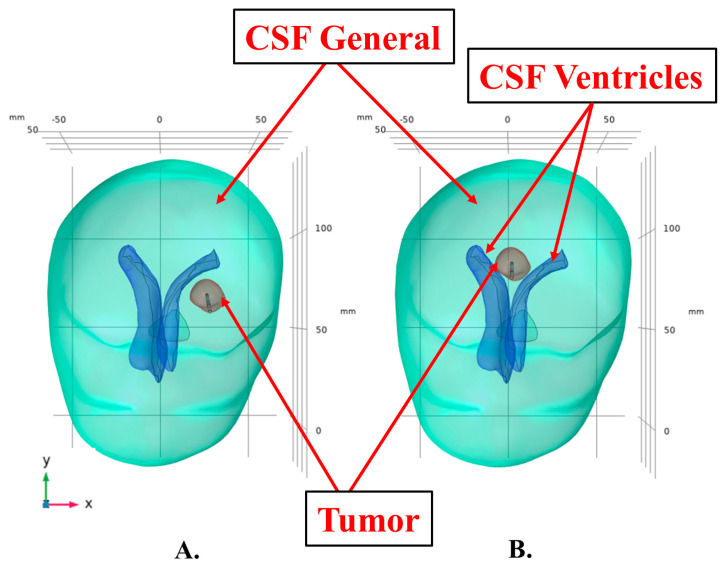
(**A**) Tumor located 3 mm away and exposed to only one CSF ventricle. (**B**) Tumor located in-between the two CSF ventricles (butterfly gliomas).

**Figure 5 cancers-15-04554-f005:**
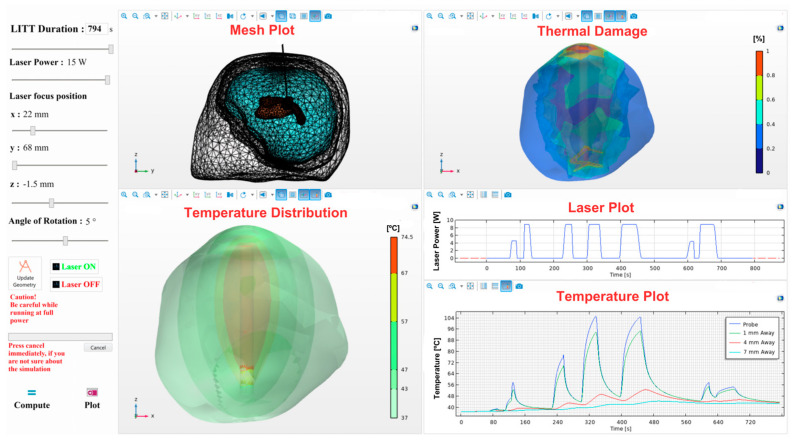
Simulation application’s final, user-friendly GUI. The user interface consists of five displays, including mesh geometry, tissue injury rate, temperature contour, laser plot, and temperature plot. The user can enter the desired surgical parameters by sliding the bars according to patient-specific data and then pressing the compute button to obtain the results. The plot icon is used to display results on the user interface.

**Figure 6 cancers-15-04554-f006:**
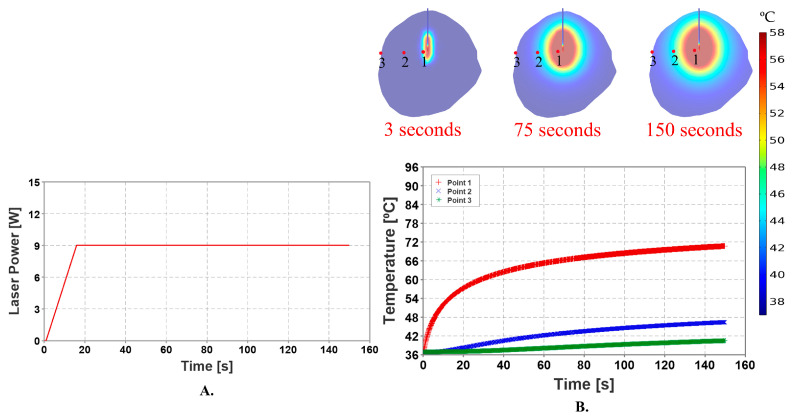
The simulation with a constant laser power application to the laser probe. (**A**) depicts the application of 9 watts of laser power to the laser probe over a duration of 150 s. (**B**) depicts three temperature maps demonstrating thermal injury to the tumor. The temperature of the measuring probe at point one, continued to rise after the application of the power pulse. After 150 s, the measured temperature was around 70.69 °C. The temperature at point two started increasing gradually since the point was placed on the tumor surface. The temperature at this point was measured at about 46.4 °C at the end of the simulation. To make sure that the model’s response matched the course of therapy, the constant laser power mechanism was modeled and verified with analytical calculations.

**Figure 7 cancers-15-04554-f007:**
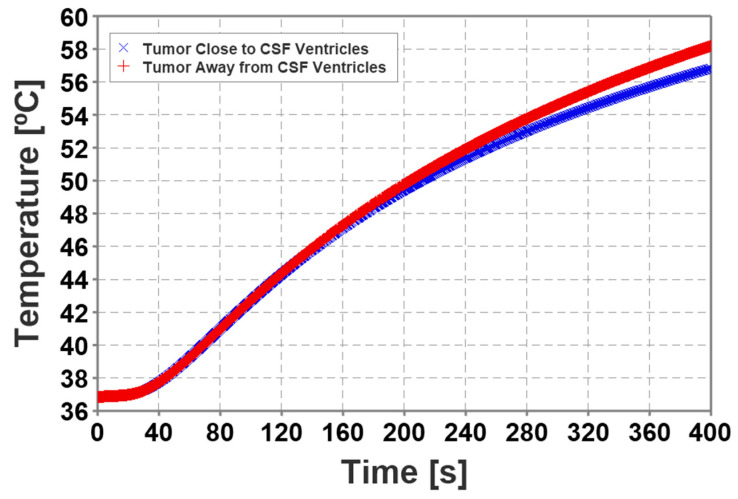
Comparison between two scenarios of the tumor location; away from the CSF ventricles and close to the CSF ventricles (Butterfly Gliomas). The location of the tumor plays a significant role in predicting thermal injury to the tumor. Since the CSF ventricles were modeled as fluids with convectively enhanced conductivity, the tumor closest to the ventricles shows significantly lower temperature than the tumor farther from the ventricles.

**Figure 8 cancers-15-04554-f008:**
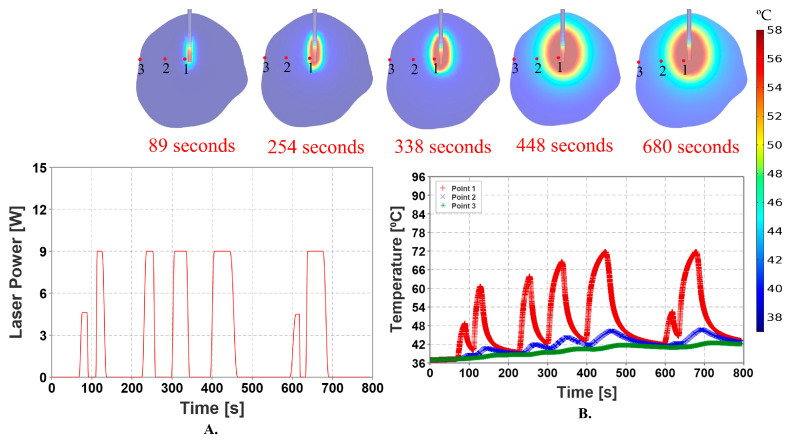
Simulation with a pulsed laser power application to the laser probe, with the laser probe held stationary throughout the simulation. (**A**) depicts the application of pulsed laser power, as determined by the data from a surgical procedure to the laser probe over the course of 794 s. Visualizing through the MRTi reveals that the first pulse is initiated after 75 s with an intensity of 30% of the maximum laser power of 15 Watts to assure the accuracy of the laser position. After confirming the laser’s position, the laser pulse is administered with a maximum power of 60% and cooling durations that are appropriate for the situation. (**B**) shows the temperature maps at different time intervals at point one, two & three. The maximum temperature achieved at point one is around 71.68 °C.

**Figure 9 cancers-15-04554-f009:**
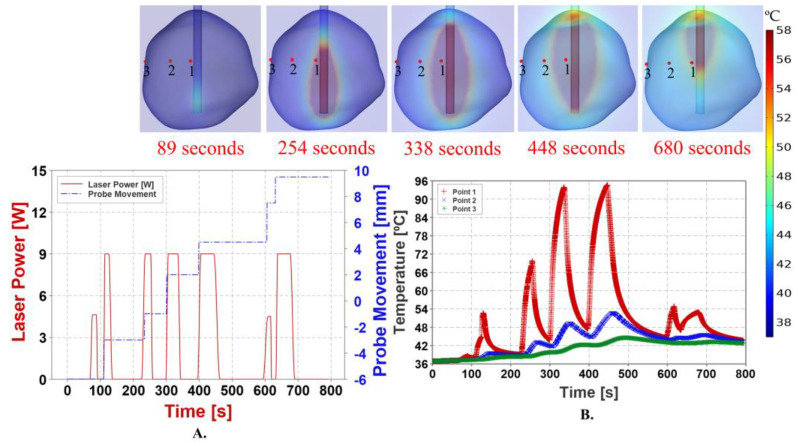
Simulation with a pulsed laser power application to the laser probe, with the laser probe continuously retracted from the distal end to the proximal end of the tumor. (**A**) depicts the application of pulsed laser power, as determined by the data from a surgical procedure to the laser probe over the course of 794 s. The laser pulse is administered with a maximum power of 60% and cooling durations that are appropriate for the situation. Figure 9B shows the temperature maps at different time intervals at point one, two & three. The maximum temperature achieved at point one is around 94.59 °C. Point three being on the surface of the tumor, the temperature at this point is below necrosis temperature of 57 °C throughout the simulation ensuring the safety of the treatment.

**Figure 10 cancers-15-04554-f010:**
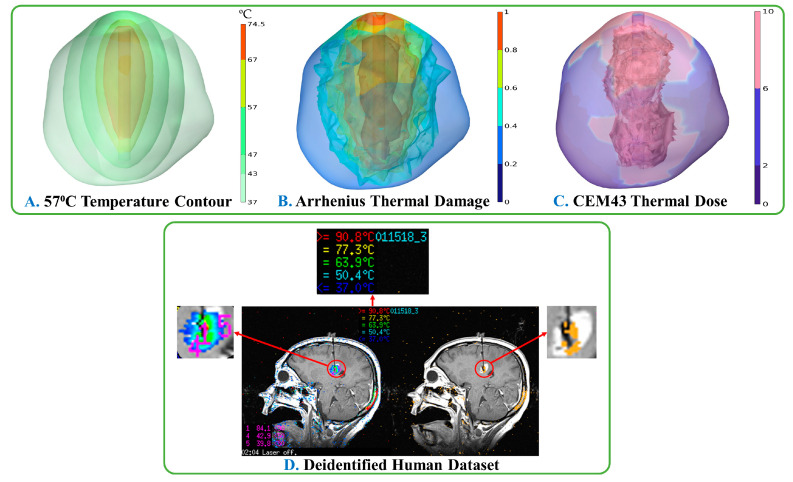
Analyzing the thermal damage caused by three different time-dependent temperature changes and thermal dose during surgery. Figure 10 indicates that the extent of thermal damage is dependent on the duration of the surgical procedure, as demonstrated by the thermal dose measurements. The temperature contour (Figure 10A) at 57 °C delineates the range of temperatures present within the lesion, spanning from 37 °C to 74.5 °C. The present study investigates the Arrhenius thermal damage model (Figure 10B) to assess the damage incurred as a result of blood perfusion over time. The thermal dose’s damage can be quantified by CEM43 (Figure 10C), which represents the cumulative equivalent minutes at 43 °C. Based on the LITT system in use, the results can be compared with the deidentified human dataset (Figure 10D).

**Figure 11 cancers-15-04554-f011:**
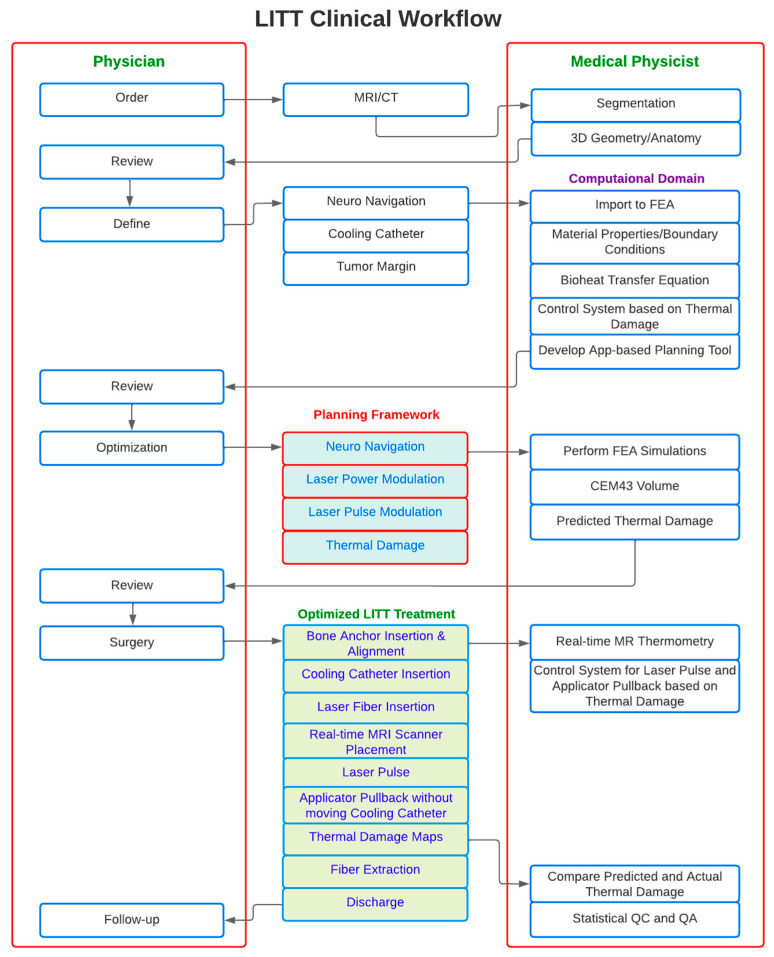
Proposed workflow for LITT treatment planning. The task carried out by the physician is described on the left side of the figure, and the task carried out by the medical physicist is described on the right side of the figure. Diagnoses, MR and CT image segmentation, modeling 3D simulations in FEA software, neuro navigation for catheter and applicator placement, laser power and pulse modulation on the app-based GUI, surgery, discharge, and post-therapy follow-up are all steps in the LITT treatment.

**Table 1 cancers-15-04554-t001:** Thermal properties used in models.

Part	Heat Capacity at Constant Pressure, $C_{p}$ [J/(kg·K)]	Density, ρ $\left[kg/m^{3}\right]$	Thermal Conductivity, k [W/(m·K)]	Dynamic Viscosity, µ [Pa·s]	Velocity, u [m/s]	Refs.
Brain (grey & white matter)	3630	1046	0.51	-	-	[32,33,34]
Tumor	3700	1056	0.57	-	-	[32,33,34]
CSF (general and ventricles)	4096	1007	0.62	7.84 × 10^−4^	0.08	[30,32]
Laser Probe	750	2200	1.38	-	-	[35]
Blood	3617	1050	0.52	3.65 × 10^−4^	1.5 × 10^−3^	[32,33,34]

**Table 2 cancers-15-04554-t002:** Values of different bioheat properties like the thermal properties of blood and blood perfusion rates of different domains [32,41,42].

Property	Value
Blood Temperature, $T_{b}$	310 [K]
Blood Perfusion in Skull, $\omega_{b\_skull}$	$10\;[\frac{mL}{min}/kg]$
Blood Perfusion in Brain, $\omega_{b\_brain}$	$559\;[\frac{mL}{min}/kg]$
Blood Perfusion in Tumor, $\omega_{b\_tumor}$	$222.3\;[\frac{mL}{min}/kg]$
Activation Energy, $E_{a}$	$2.38\times 10^{5}$ [J/mol]
Frequency Factor, A	$1.8\times 10^{36}$ [1/s]

**Table 3 cancers-15-04554-t003:** Cerebrospinal Fluid properties and calculation for Nusselt Number [30,32,33,34,46,47,48].

Thermal Conductivity, k [W/(m·K)]	Heat Capacity at Constant Pressure, $C_{p}$ [J/(kg·K)]	Dynamic Viscosity, µ [Pa·s]	Prandtl Number, Pr	Reynolds Number, Re	Grashof Number, Gr	Nusselt Number, Nu
0.62	4096	7.84 × 10^−4^	5.179	285	55.02	1.725

## Data Availability

Research data are stored in an institutional repository and will be shared with the corresponding author upon request.

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
