# Peer review of "Development of a Treatment Planning Framework for Laser Interstitial Thermal Therapy (LITT)"

_cancers, 2023, doi:10.3390/cancers15184554_

Round 1

Reviewer 1 Report

It will be about the description and utilization of LITT in the treatment of GBM so as a study and technical note I think it can be okay, however, I think it is more crucial than ever to make focus on the two most important notions of this project namely: is there the space for the recovery of material to do biopsy? but more importantly how can a study be planned by looking at progression free survival and overall survival in consideration of the molecular parameters of gbm?

Author Response

We, the authors, thank the reviewer for taking time to read our manuscript and to provide suggestions for its improvement. Below, we provide point-by-point responses to reviewer comments. Changes to manuscript text are in red font.

Comment 1:

It will be about the description and utilization of LITT in the treatment of GBM so as a study and technical note I think it can be okay, however, I think it is more crucial than ever to make focus on the two most important notions of this project namely: is there the space for the recovery of material to do biopsy? but more importantly how can a study be planned by looking at progression free survival and overall survival in consideration of the molecular parameters of gbm?

Response: We are in agreement with the Reviewer that this manuscript is intended to serve as a technical note for development of a LITT planning framework. The Reviewer does bring up two interesting questions. 1) We believe the workflow for LITT can seamlessly be integrated with obtaining tumor tissue for histologic confirmation and/or molecular analysis. 2) While in an ideal world a study might be designed to assess the effectiveness of such a LITT planning framework on progression-free and/or overall survival, this is likely not practical. As the Reviewer notes, the molecular heterogeneity of glioblastoma and the presumptive incremental improvement in ablation will render a robust clinical study unrealistic as the number of patients would be prohibitively high.

Reviewer 2 Report

1.     The novelty of the work must be clearly addressed and discussed, compare your research with existing research findings and highlight novelty (compare your work with existing research findings and highlight novelty),

2.     The main objective of the work must be written on the more clear and more concise way at the end of introduction section, discuss necessity for the conducted review work, 

3.     Introduction section must be written on more quality way, i.e. more up-to-date references addressed. Research gap should be delivered on more clear way with directed necessity for the conducted research work. To enrich the introduction section, the following published papers can be cited:

4.     Conclusion section is missing some perspective related to the future research work, quantify main research findings

5.     English language should be carefully checked and carefully check paper for language typos,

6.     Please provide section related to the review methodology to clarify how review was organized, how references were selected and what are the possible limitations of the work,

7.     Add in the abstract section data related to the main research outcomes.

Author Response

We, the authors, thank the reviewer for taking time to read our manuscript and to provide suggestions for its improvement. Below, we provide point-by-point responses to reviewer comments. Changes to manuscript text are in red font.

Comments and Suggestions for Authors

Comment 1: The novelty of the work must be clearly addressed and discussed, compare your research with existing research findings and highlight novelty (compare your work with existing research findings and highlight novelty),

Response: We appreciate the helpful comment. The introduction (pg. no.2) and discussion (pg. no.15) sections of the manuscript has been edited to address the novelty of study. The proposed workflow for LITT treatment framework has been moved to the discussion section which was previously misplaced in the methods section.

Comment 2:  The main objective of the work must be written on the more clear and more concise way at the end of introduction section, discuss necessity for the conducted review work,

Response: We thank the reviewer for pointing out our lack of clarity. The introduction section (pg. no.3) has been appropriately modified.

Comment 3:  Introduction section must be written on more quality way, i.e., more up-to-date references addressed. Research gap should be delivered on more clear way with directed necessity for the conducted research work. To enrich the introduction section, the following published papers can be cited:

Response: We thank the reviewer for pointing out our oversight. We have included a few selected additional references in the introduction section (pg. no.2) that have been published within last five years. As our manuscript is not a review or meta-analysis paper, we have included only a few of these (review and original works) that have particular relevance. We acknowledge some may have been excluded but this is unavoidable to maintain reader focus.

Comment 4:  Conclusion section is missing some perspective related to the future research work, quantify main research findings.

Response: We appreciate the suggestion. We modified the conclusion section (pg. no.17) to quantify main research findings and provide perspective for future work.

Comment 5:  English language should be carefully checked and carefully check paper for language typos.

Response: We thank the reviewer for pointing out our oversight. We have thoroughly reviewed and fixed grammatical errors and any typos.

Comment 6:  Please provide section related to the review methodology to clarify how review was organized, how references were selected and what are the possible limitations of the work,

Response: We appreciate the suggestion. This is a LITT treatment planning framework development manuscript. Our review of the literature was limited and focused only to provide readers with a snapshot of current state of art as context. The reviewer’s suggestion to provide a comprehensive review is of interest, and one the authors are now seriously considering. We thank the reviewer for the suggestion.

Comment 7:  Add in the abstract section data related to the main research outcomes.

Response: We thank the reviewer for the helpful suggestion. We have added information related to the main research outcomes in the abstract section (pg. no.1).

We thank the reviewer for the comments and identifying our oversights to help us improve the reader’s experience.

Reviewer 3 Report

Reviewer’s comments

The authors conducted a study to develop a treatment planning tool for Laser Interstitial Thermal Therapy (LITT). The concept is in line with other thermal treatment planning tools such as Hyperthermia Treatment Planning (HTP). The study is interesting, however, there are areas in the manuscript that require further improvement. The authors have shown a few treatment parameters, then talk about GUI development and some MR thermometry clinical data. The transition to LITT is not well established in the paper and it’s confusing. Either the authors need to focus on the patient-specific treatment approach to establish optimum treatment parameters or focus on developing the tool and integrating it with MR thermometry with a clinical example. A good reference for that is the study by H.P. Kok et al (http://dx.doi.org/10.1080/02656736.2017.1295323). The reviewer comments are described below.

Major comments:

1.      The aim of the paper is to develop a tool for LITT, and the authors used patient-specific model to establish that. One major drawback of the paper is authors have shown some treatment approaches, but the optimum treatment parameters are not well established. Is there a prior work of the authors establishing those? The idea of developing tools while looking at the treatment parameters is not so comprehensive from the reviewer’s point of view.  

2.      The drawback of modeling is the lack of physics related to laser-tissue interaction. Instead, a mathematical equation is used to calculate the heat source. The reviewer has concerns about the energy being delivered from the laser probe as the distribution may be affected by ‘d’ (assuming the beam diameter). The authors have shown modulated power deposition as well which is impacted by input power as well. It may not have been an issue if the study is focused only on determining treatment parameters, since the approach is used to integrate as a clinical tool it needs more justification. How can the authors validate the heat source is modeled accurately? Is there model calibration data in gel phantom or MR thermometry data comparing the simulation vs experiment?

3.      How different is the cylindrical probe compared to the actual probe used in clinics? How is the tip of the probe modeled it looks like a flat cylindrical probe. Doesn’t the surface area of the tip affect the heating rates, it may be critical in pulsed heating.

4.      Is the GUI developed integrated with COMSOL? Explain more on what is the capability of GUI. What needs to be done initially in COMSOL and what can be done only using GUI?

5.      The pull back mechanism is modelled using a piecewise function. Is the function used only for the heat source? In Fig 10 it looks like the probe position is dynamic. Is it a static or dynamic meshing approach? Can the authors add more details about how the modeling approach is established including capturing tissue damage for pull back mechanism?

6.      In equation 2 the authors assumed beta = 0 which results in rh0*C*omega*deltaT, conventional form of perfusion term. Why do the authors want to emphasize the equation with beta if it’s not relevant? It is even repeated in equation 3. The reviewer would suggest modifying the equation if it doesn’t have a value.

7.      Why do the authors want to quantify tissue damage in both Arrhenius and CEM43 equations? CEM43 seems to be widely used over the Arrhenius method, even in clinical devices.  What the authors mentioned about perfusion elevation may also be incorporated into the CEM43 approach as well by considering temperature elevated perfusion in tissues. Unless there is a proper justification or application in clinal treatment the reviewer suggests using only the CEM43 way of calculating thermal dose in the manuscript as the paper is intended to develop a tool for clinical use.

8. In section 2.1.0 line 238, the authors mentioned fiber optic sensors used for temperature measurement. It is not clear which data are measured using fiber optic sensors. Also, the authors mentioned experiments several times in the manuscript, and which are all the experimental data are not clear though out the manuscript.

9.      The authors also mentioned obtaining clinical data and comparing it with MR thermometry. Did the authors adopt the clinical treatment parameters to perform the simulations or vice versa? Is the pull back probe mechanism from clinical data? The conceptualization for choosing the treatment parameters and the data shown as clinical data is not very clear.

10.   The unit of perfusion used is [1/s] which is not in SI units and may not be easily understandable to the readers unless do the conversion with density. Add the values of perfusion units in a much more conventional way according to how the density term is considered in the perfusion term. It seems like the unit is in kg/m^3/s or in ml/min/kg.

Minor comments

11.   Conclusion doesn’t justify the title of the paper it needs to be restated.

12.   Include the abbreviation of CSF in the text.

13.   Hard to read the labels in Fig 1 and others. Increase the font size of figure labels.

14.   In which CAD format the model is imported to COMSOL? How easy is the adoption of the CAD format from DICOM images when it is applied as a tool in clinics?

15.   Add the description of ‘d’ in equation 14.

Author Response

We, the authors, thank the reviewer for taking time to read our manuscript and to provide suggestions for its improvement. Below, we provide point-by-point responses to reviewer comments. Changes to manuscript text are in red font.

Comments and Suggestions for Authors

The authors conducted a study to develop a treatment planning tool for Laser Interstitial Thermal Therapy (LITT). The concept is in line with other thermal treatment planning tools such as Hyperthermia Treatment Planning (HTP). The study is interesting, however, there are areas in the manuscript that require further improvement. The authors have shown a few treatment parameters, then talk about GUI development and some MR thermometry clinical data. The transition to LITT is not well established in the paper and it’s confusing. Either the authors need to focus on the patient-specific treatment approach to establish optimum treatment parameters or focus on developing the tool and integrating it with MR thermometry with a clinical example. A good reference for that is the study by H.P. Kok et al (http://dx.doi.org/10.1080/02656736.2017.1295323). The reviewer comments are described below.

Major Comments:

Comment 1: The aim of the paper is to develop a tool for LITT, and the authors used patient-specific model to establish that. One major drawback of the paper is authors have shown some treatment approaches, but the optimum treatment parameters are not well established. Is there a prior work of the authors establishing those? The idea of developing tools while looking at the treatment parameters is not so comprehensive from the reviewer’s point of view.

Response: We thank the reviewer for pointing out our oversight. We have developed a framework for the tool. The objective of this manuscript was to provide readers with the description of the framework and its development, to enable readers to evaluate our approach. Future studies will focus on identifying patient-specific optimum treatment parameters.

Comment 2:  The drawback of modeling is the lack of physics related to laser-tissue interaction. Instead, a mathematical equation is used to calculate the heat source. The reviewer has concerns about the energy being delivered from the laser probe as the distribution may be affected by ‘d’ (assuming the beam diameter). The authors have shown modulated power deposition as well which is impacted by input power as well. It may not have been an issue if the study is focused only on determining treatment parameters, since the approach is used to integrate as a clinical tool it needs more justification. How can the authors validate the heat source is modeled accurately? Is there model calibration data in gel phantom or MR thermometry data comparing the simulation vs experiment?

Response: We thank the reviewer for pointing out our lack of clarity.

  1. Authors agree that the current model doesn’t capture laser-tissue interaction dynamics. However, authors believe that the heat source approximation based upon the laser probe tip will not be a significant factor as the goal of the treatment is maximum tumor coverage based on metrics such as T>57ºC, Arrhenius Damage and CEM43 T10. Heat transfer (conduction and convection) from the heat source, i.e., laser probe tip and the deposited beam diameter ‘d’ to the tumor boundary is more significant factor (d, mm << tumor diameter, cm).
  2. Authors agree that using a constant beam diameter ‘d’ for modulated power is a limitation and does not capture all the power modulation dynamics. To accommodate multiple laser probe tips from manufacturers (Monteris Neuroblate SideFire and FullFire, Medtronics Visualase Cylindrical) and different cooling strategies, we plan to use experimental probe specific power distribution profiles in future studies. This will also include the variations in the distribution profiles corresponding to the power modulation. Appropriate text has been added to the manuscript.
  3. A clear statement has been made in the abstract (pg. no.1) and introduction (pg. no.2) to address the confusion in the scope of this study. This study focuses on developing a robust framework for LITT treatment planning for multiple manufacturers.
  4. This paper is about developing the framework for the LITT treatment planning tool. To clarify this, we added a transition paragraph in the discussion before the text related to the discussions on approaches and issues related to clinical translation of the developed framework.
  5. Model verification and validation in gel phantoms with thermometry data are part of the future studies.

The manuscript has been modified to reflect the above-mentioned points. Authors thank the reviewer for his insightful comments. Authors believe these modifications based upon reviewer’s comments significantly improved the quality and readability of the manuscript.

Comment 3:  How different is the cylindrical probe compared to the actual probe used in clinics? How is the tip of the probe modeled it looks like a flat cylindrical probe. Doesn’t the surface area of the tip affect the heating rates, it may be critical in pulsed heating.

Response: We thank the reviewer for pointing out our lack of clarity. As mentioned in the response to the previous comment, the cylindrical probe serves as a generic laser tip for the developed framework. Future improvements will include manufacturer and probe style specific laser power distribution profile.

Comment 4:  Is the GUI developed integrated with COMSOL? Explain more on what is the capability of GUI. What needs to be done initially in COMSOL and what can be done only using GUI?

Response: We thank the reviewer for the comment. Yes, the GUI is integrated with COMSOL. COMSOL has a module to deploy the developed application to mobile platforms such as iPads. Neurosurgeons can adjust the provided parameters on the screen to plan the treatment. In future studies, we plan to perform GUI testing and improve the usability, add or remove parameters from the GUI.

Comment 5:  The pull back mechanism is modelled using a piecewise function. Is the function used only for the heat source? In Fig 10 it looks like the probe position is dynamic. Is it a static or dynamic meshing approach? Can the authors add more details about how the modeling approach is established including capturing tissue damage for pull back mechanism?

Response: We thank the reviewer for pointing out our oversight. The heat source was modeled using a piecewise function, with the laser probe modeled using static meshing approach in the FEA software within the tumor. The temperature measuring point probes were modeled with static approach in COMSOL. The results of the pullback heat source were captured with the animation plots in COMSOL. Details of the modelling approach have been added to the manuscript.

Comment 6:  In equation 2 the authors assumed beta = 0 which results in rh0*C*omega*deltaT, conventional form of perfusion term. Why do the authors want to emphasize the equation with beta if it’s not relevant? It is even repeated in equation 3. The reviewer would suggest modifying the equation if it doesn’t have a value.

Response: We thank the reviewer for the helpful suggestion. We have modified the equation (pg. no.6) accordingly.

Comment 7:  Why do the authors want to quantify tissue damage in both Arrhenius and CEM43 equations? CEM43 seems to be widely used over the Arrhenius method, even in clinical devices.  What the authors mentioned about perfusion elevation may also be incorporated into the CEM43 approach as well by considering temperature elevated perfusion in tissues. Unless there is a proper justification or application in clinal treatment the reviewer suggests using only the CEM43 way of calculating thermal dose in the manuscript as the paper is intended to develop a tool for clinical use.

Response: We thank the reviewer for the helpful suggestion. Monteris uses CEM43, T>57ºC and Medtronic uses Arrhenius Damage in their clinical system [1,2,3]. Based upon the neurosurgeon’s preference and the systems being used, they can pick and choose the appropriate approach for modelling thermal damage.

Comment 8:  In section 2.1.0 line 238, the authors mentioned fiber optic sensors used for temperature measurement. It is not clear which data are measured using fiber optic sensors. Also, the authors mentioned experiments several times in the manuscript, and which are all the experimental data are not clear though out the manuscript.

Response: We thank the reviewer for pointing out our oversight. We apologize for the confusion. The fiber optic sensors correspond to the temperature measuring point probes in COMSOL, no fiber optic sensors were used and changes to the text (pg. no.8) have been made accordingly. Also, the experiments resemble the simulations and not the phantom experiments. The word ‘experiment’ has been changed to simulations in the manuscript.

Comment 9:  The authors also mentioned obtaining clinical data and comparing it with MR thermometry. Did the authors adopt the clinical treatment parameters to perform the simulations or vice versa? Is the pull back probe mechanism from clinical data? The conceptualization for choosing the treatment parameters and the data shown as clinical data is not very clear.

Response: We thank the reviewer for pointing out our oversight. We obtained a gif file of a laser log from a LITT surgery which helped us in modeling approximate pullback mechanism of the heat source. For verification and validation of the GUI based treatment planning framework, it is necessary to obtain patient specific clinical data from the LITT systems, which will be demonstrated in our future work.

Comment 10:  The unit of perfusion used is [1/s] which is not in SI units and may not be easily understandable to the readers unless do the conversion with density. Add the values of perfusion units in a much more conventional way according to how the density term is considered in the perfusion term. It seems like the unit is in kg/m^3/s or in ml/min/kg.

Response: We thank the reviewer for pointing out our oversight. The blood perfusion values in Table 2. (pg. no.7) have been converted to ml/min/kg.

Minor Comments:

Comment 11:  Conclusion doesn’t justify the title of the paper it needs to be restated.

Response: We thank the reviewer for pointing out our oversight. The conclusion section (pg. no.17) is modified to justify the title of the paper.

Comment 12:  Include the abbreviation of CSF in the text.

Response: We thank the reviewer for pointing out our oversight. Appropriate changes have been made to the text in the abstract section (pg. no.1).

Comment 13:  Hard to read the labels in Fig 1 and others. Increase the font size of figure labels.

Response: We thank the reviewer for pointing out our oversight. The figure labels in the manuscript have been improved to account for readability.

Comment 14:  In which CAD format the model is imported to COMSOL? How easy is the adoption of the CAD format from DICOM images when it is applied as a tool in clinics?

Response: We thank the reviewer for pointing out our oversight. The DICOM images are converted to STL files in the MATERIALISE software and imported into COMSOL. The image processing needs to be carried out by the medical physics team in consultation with the Neurosurgeons to identify accurate boundaries of the domains.

Comment 15:  Add the description of ‘d’ in equation 14.

Response: We thank the reviewer for pointing out our oversight. An appropriate description has been added to the text (pg. no.8).

We thank the reviewer for the comments and identifying our oversights to help us improve the reader’s experience.

References:

  1. Patel, Nitesh V., et al. "Laser interstitial thermal therapy technology, physics of magnetic resonance imaging thermometry, and technical considerations for proper catheter placement during magnetic resonance imaging–guided laser interstitial thermal therapy." Neurosurgery 79 (2016): S8-S16.
  2. Patel, Nitesh V., Kiersten Frenchu, and Shabbar F. Danish. "Does the thermal damage estimate correlate with the magnetic resonance imaging predicted ablation size after laser interstitial thermal therapy?." Operative Neurosurgery 15.2 (2018): 179-183.
  3. Mohammadi, Alireza M., and Jason L. Schroeder. "Laser interstitial thermal therapy in treatment of brain tumors–the Neu-roBlate System." Expert review of medical devices 11.2 (2014): 109-119.

Round 2

Reviewer 2 Report

Accept in current form

Author Response

Response to reviewer comments:

We, the authors thank the reviewer for taking time to read our manuscript and to provide suggestions for its improvement in round 2 revisions. Below, we provide point-by-point responses to reviewer comments. Changes to manuscript figure numbers are in red font.

Comments and Suggestions for Authors

Comment 1: Accept in current form.

Response: We thank the reviewer for accepting the manuscript for publication.

Reviewer 3 Report

The authors have addressed all the concerns raised by the reviewer. The drawbacks of the paper have been addressed and future plans are discussed. Overall, the manuscript looks comprehensive now. The reviewer recommends the publication of the paper after addressing minor comments below. For the betterment of readers, the reviewer has a few comments to improve the presentation quality of the figures used in the manuscript.
1) Figure 6- Increase the font size and range for better visualization. Add scale for temperature contour in B
2) Figure 7- Adjust the range for the y-axis for better visualization.
3) Figure 8 B) - Adjust the temperature range and font size for better visualization (y-axis).

4) Figure 9 (B)- Comment same as 3
5) Improve font size in the above-mentioned figures.

Author Response

Response to reviewer comments:

We, the authors thank the reviewer for taking time to read our manuscript and to provide suggestions for its improvement in round 2 revisions. Below, we provide point-by-point responses to reviewer comments. Changes to manuscript figure numbers are in red font.

Comments and Suggestions for Authors

The authors have addressed all the concerns raised by the reviewer. The drawbacks of the paper have been addressed and future plans are discussed. Overall, the manuscript looks comprehensive now. The reviewer recommends the publication of the paper after addressing minor comments below. For the betterment of readers, the reviewer has a few comments to improve the presentation quality of the figures used in the manuscript.

Minor Comments:

Comment 1: Figure 6- Increase the font size and range for better visualization. Add scale for temperature contour in B

Response: We thank the reviewer for pointing out our oversight. We have increased the font size and range for the figure. Also, we have added a scale for temperature contour in Figure 6.

Comment 2:  Figure 7- Adjust the range for the y-axis for better visualization.

Response: We thank the reviewer for the comment. Appropriate changes have been made to the range (y-axis) in Figure 7.

Comment 3:  Figure 8 B) - Adjust the temperature range and font size for better visualization (y-axis).

Response: We thank the reviewer for pointing out our oversight. Changes to Figure 8 have been made accordingly.

Comment 4:  Figure 9 (B)- Comment same as 3

Response: We thank the reviewer for the comment. Changes in the temperature range on y-axis have been made and Figure 9 has been updated accordingly.

Comment 5: Improve font size in the above-mentioned figures.

Response: We thank the reviewer for pointing out our oversight. Font size in all the figures mentioned above has been increased.